# Systematic Review on Biosynthesis of Silver Nanoparticles and Antibacterial Activities: Application and Theoretical Perspectives

**DOI:** 10.3390/molecules26165057

**Published:** 2021-08-20

**Authors:** Shafqat Qamer, Muhammad Hibatullah Romli, Fahrudin Che-Hamzah, Norashiqin Misni, Narcisse M. S. Joseph, Nagi A. AL-Haj, Syafinaz Amin-Nordin

**Affiliations:** 1Department of Medical Microbiology, Faculty of Medicine and Health Sciences, Universiti Putra Malaysia, 43400 Serdang, Selangor, Malaysia; gs56387@student.upm.edu.my (S.Q.); norashiqin@upm.edu.my (N.M.); narcissems@upm.edu.my (N.M.S.J.); 2Department of Basic Medical Science, College of Medicine, Prince Sattam Bin Abdulaziz University, Alkharj 11942, Saudi Arabia; 3Department of Rehabilitation Medicine, Faculty of Medicine and Health Sciences, Universiti Putra Malaysia, 43400 Serdang, Selangor, Malaysia; mhibatullah@upm.edu.my; 4Malaysian Research Institute on Ageing (MyAgeing), Universiti Putra Malaysia, 43400 Serdang, Selangor, Malaysia; 5Orthopaedic Department, Faculty of Medicine and Health Sciences, Universiti Putra Malaysia, 43400 Serdang, Selangor, Malaysia; fahrudinch@upm.edu.my; 6Department of Medical Microbiology, Faculty of Medicine and Health Sciences, Sana’a University, Sana’a 009671, Yemen; naji2005@gmail.com

**Keywords:** data analytics, silver nanoparticles, biosynthesis, antibacterial activities

## Abstract

The biosynthesis of silver nanoparticles and the antibacterial activities has provided enormous data on populations, geographical areas, and experiments with bio silver nanoparticles’ antibacterial operation. Several peer-reviewed publications have discussed various aspects of this subject field over the last generation. However, there is an absence of a detailed and structured framework that can represent the research domain on this topic. This paper attempts to evaluate current articles mainly on the biosynthesis of nanoparticles or antibacterial activities utilizing the scientific methodology of big data analytics. A comprehensive study was done using multiple databases—Medline, Scopus, and Web of Sciences through PRISMA (i.e., Preferred Reporting Items for Systematic Reviews and Meta-Analyses). The keywords used included ‘biosynthesis silver nano particles’ OR ‘silver nanoparticles’ OR ‘biosynthesis’ AND ‘antibacterial behavior’ OR ‘anti-microbial opposition’ AND ‘systematic analysis,’ by using MeSH (Medical Subject Headings) terms, Boolean operator’s parenthesis, or truncations as required. Since their effectiveness is dependent on particle size or initial concentration, it necessitates more research. Understanding the field of silver nanoparticle biosynthesis and antibacterial activity in Gulf areas and most Asian countries also necessitates its use of human-generated data. Furthermore, the need for this work has been highlighted by the lack of predictive modeling in this field and a need to combine specific domain expertise. Studies eligible for such a review were determined by certain inclusion and exclusion criteria. This study contributes to the existence of theoretical and analytical studies in this domain. After testing as per inclusion criteria, seven in vitro studies were selected out of 28 studies. Findings reveal that silver nanoparticles have different degrees of antimicrobial activity based on numerous factors. Limitations of the study include studies with low to moderate risks of bias and antimicrobial effects of silver nanoparticles. The study also reveals the possible use of silver nanoparticles as antibacterial irrigants using various methods, including a qualitative evaluation of knowledge and a comprehensive collection and interpretation of scientific studies.

## 1. Introduction

Silver nanoparticles Ag-NPs have shown exceptional properties. There are many applications of the Ag-NPs in bio-medical fields. The most significant advantage of Ag-NPs is the production of antibiotics. Currently, biosynthesized Ag-NPs have been extensively studied in the last ten years [1]. The size of Ag-NPs is adjusted according to the specific application [2]. For instance, the Ag-NPs that are prepared for drug delivery are greater than 100 nm in size. Furthermore, Ag-NPs are significantly used in antimicrobial applications since they have shown antimicrobial properties. These unique characteristics of silver nanoparticles enabled their use in the fields of nanomedicine, pharmacy, biosensing, and biomedical engineering. 

Ag-NPs have many practical applications, such as antibacterial and anticancer therapeutics, diagnostics, and optoelectronics, water disinfection, and other clinical/pharmaceutical applications. Silver is known to have fascinating properties. In addition, they are cost-effective, are abundantly found in nature, and silver nanoparticles have undeniable potential compared with stable gold nanoparticles [2]. Currently, antibacterial resistance has become very frequent and is considered as the most complex and global health challenge. There is an urgent need to discover and synthesize new biomedicines or find natural alternatives, as it is of utmost priority and high demand in the present circumstances. Globally, it has become the most challenging situation to deal with the hospital and community-acquired infections that are particularly caused by multidrug-resistant bacteria [3]. Moreover, the death rate has been significantly increasing due to these multidrug-resistant bacterial pathogens. Multiple studies suggest that the extensive and improper use of antibiotics resulted in the formation of new multidrug bacterial resistance genes [3]. Therefore, it is essential to reduce the extensive intake of antibiotics. It is important to discover and develop novel warfare schemes to fight against multidrug-resistant bacteria and reduce chemical drug intake complications to treat microbial infections.

Antibacterial materials are primarily divided into inorganic, organic, and natural materials. Natural antibacterial materials have a limited application range, while organic antibacterial materials with low heat resistance are prone to causing bacterial drug resistance. In terms of durability, heat resistance, and the emergence of antibiotic resistance in bacterial strains, inorganic antibacterial materials outperform organic antibacterial materials. As a result, it has received a lot of attention in past years [4]. For example, Ag+ ions are a broad-spectrum antibacterial agent that may effectively stop bacteria, fungi, and algae from growing [5]. Many inorganic nanoparticles (NPs) have shown severe cytotoxicity, indicating that a new generation of bactericidal materials could be developed [6]. Silver NPs, specifically, have been intensively researched due to their great antibacterial action while causing minimum disruption to human cells [7]. For example, silicon nanowires coated with silver nanoparticles exhibit effective antibacterial action [8]. However, most studies [9,10] have concentrated on the antibacterial performance and mechanisms of Ag-NPs, with little effort devoted to the development of environmentally friendly and biocompatible preparation techniques. Ag-NPs have different physicochemical and biological properties than their counterparts related to the higher surface to volume ratio [11].

Multiple pieces of research have been conducted to demonstrate the impact of shapes of Ag-NPs on their antibacterial activity. The number and position of surface plasmon resonance (SPR) peaks are dependent on the shape of Ag-NPs. For example, spherical particles show a single scattering peak. In contrast, anisotropic shapes such as rods, triangular prisms, and cubes show multiple scattering peaks in the visible wavelengths due to highly localized charge polarizations at corners and edges [12,13]. As a result, achieving a size-tunable synthesis of Ag-NPs with substantial surface area and surface activity and poor stability, and a strong aggregation potential is a considerable problem [14,15]. Several forms of Ag-incorporated nanomaterials have been discovered, and Ag-NPs immobilized on various inorganic and organic substrates have been found to have increased and extended antibacterial properties [16].

At present, it is still unclear how silver inhibits bacterial growth through its antibacterial properties [17]. According to some researchers, the antibacterial impact of Ag-NPs could be explained by three different hypotheses: contact action, the formation of reactive oxygen species (ROS), and release of Ag+. For example Morones [18] claims that Ag-NPs can be linked to the surface of a bacteria’s cell membrane to disrupt the cell membrane’s functions and enter through the bacteria to induce cytoplasm leakage and eventually result in the bacteria’s death. The antibacterial activity of Ag-NPs, according to Wang [19], is due to an increase in ROS concentration. They believe that ROS causes bacterial mortality by causing intracellular oxidation, membrane potential changes, and the release of cellular contents [20,21].

Ag nanoparticles (NPs) are gaining more popularity in a variety of industries, in health (medication, medical tools, pharmacology, biotechnology) as well as non-health (textiles, food, consumer goods, telecommunications, technology, power, magnetism, and environmental remediation) sectors [22,23]. Mittal discovered that synthesizing metallic nanoparticles with plant extracts was cost-effective and environment friendly and supported various analytical techniques [24]. Alongside the advantages, it is necessary to point out the problems associated with the green synthesis of Ag nanoparticles. Some major drawbacks of using plant extracts for synthesizing Ag nanoparticles include the slow rate of synthesis, the limited number of sizes and shapes of nanoparticles; and compared to conventional methods, plants produce a low yield of secreted proteins. Plants cannot be manipulated due to the choice of nanoparticles either through optimized synthesis or through genetic engineering [25].

Antimicrobial, anti-rheumatic, antiviral, bacterial, diuretic, expectorant, hypertensive, and insecticide activities of many natural products have been published in many types of research. Biogenic nanoparticle synthesis using plant extracts aiming for antimicrobial potency may synergistically impact on organisms [26]. In other words, the biological synthesis of silver nanoparticles has many promising aspects, varying by productivity and by ease of processing to become automatically secure with such a single-phase method for large-scale nanoparticle synthesis. 

There are very few documented overviews on Ag-NPs biosynthesis and antibacterial activity. The size, shape, and surface morphology of Ag-NPs are important factors in determining their characteristics. The antibacterial activities of Ag-NPs are linked to the oxidation and release of Ag+ ions into the environment, making it an excellent antibacterial agent [27]. Compared to pure silver metal, Ag-NPs are projected to have a high surface area to volume ratio and a high fraction of surface atoms, resulting in high antibacterial activity [28]. Additionally, the smaller size of Ag-NPs eases the penetration through the cell membrane and alters the intercellular processes of the bacterial cell. Most Ag nanoparticles prepared by green synthesis are investigated for their antibacterial properties or for cancer treatment. Recent studies have shown that Ag-NPs with diameters ranging from 38 to 72 nm and 17 to 29 nm may be synthesized using *Chrysanthemum indicum* [29] or *Acacia leucophloea* extract [30]. Both samples had excellent antibacterial effects. Similarly, *Ganoderma neojaponicum Imazeki* was utilized to synthesize Ag- -NPs, which could be used as chemotherapeutics against breast cancer cells [31].

The absence of predictive analytics in this field and the need to incorporate domain-specific expert knowledge highlights the current study’s need. However, a fragmented view of a theoretical and functional implementation of silver nanoparticle biosynthesis and antibacterial activities leads to various conclusions. As a result, the present study aims to know the progress of biogenic Ag-NPs. One of the objectives of this study was to look into the most advanced methods used in the biosynthesis of nanoparticles and also how they can be used as a possible treatment drug.

## 2. Methodology

Several literature reviews about the biosynthesis of metallic nanoparticles with antimicrobial properties were searched and collected. Many reports were found on the subject, and a detailed review was essential to combine all the results of the reports to drive a conclusion and avoid any conflict of information, ambiguity, and misunderstanding. This overview, which aims to highlight and classify bio-silver nanoparticles’ antibacterial activity, focuses exclusively on aggregating all such systematic reviews, as recommended by Cooper and Koenka [32]. PRISMA (i.e., Preferred Reporting Items for Systematic Reviews and Meta-Analyses) checklist was adopted to document this summary. 

### 2.1. Step 1: Formulating the Problem

In the first step, the problem was formulated about how the biogenic Ag-NP was assessed and established during examination of the antibacterial activity. This research question was further elaborated to examine whether Ag-NP could serve as smart weapons against multiple drug-resistant microorganisms, and replace the antibiotics, or if these Ag-NP could be used as a possible treatment drug to act as an effective antimicrobial agent.

### 2.2. Step 2: Literature Searches for Research Syntheses

In the second step, a systematic literature review was initially conducted in December 2019. A review of the related literature was carried out on three electronic databases, namely ScienceDirect, MEDLINE, and Scopus. A blend of specific terms was used in the research study, taken from the title, abstract, and keywords such as (‘biosynthesis silver nanoparticles’ OR ‘silver nanoparticles’ OR ‘biosynthesis’ OR ‘green synthesis’) AND (‘antibacterial activity’ OR ‘antimicrobial resistance’) AND (‘systematic review’), through the use of MeSH (Medical Subject Headings) terms, Boolean operators, parentheses, and insertions when necessary. During the searching process, a search chain or query links were created for each database. It was necessary to vary the syntax of each search criterion or query according to the database. Operational definitions of each keyword were also found in the selected articles. The final analysis included articles that contained facts, summaries, and reviewers’ remarks.

The English language was the only language that was selected for the literature search. Inclusion and exclusion criteria were defined for both the phases of the search procedure: The inclusion criteria consisted of the following: (i) articles related to biosynthesized silver nanoparticles and antibacterial effects, (ii) experimental study design, (iii) in vitro studies, and (iv) published in English. The exclusion criteria dealt with (i) reports that did not perform a systematic search (literature review), (ii) were not peer-reviewed, (iii) were not centered on biosynthesis, (iv) were not linked to antibacterial-dependent outcomes, and (v) were not published in English. Prior to the selection process, duplicate papers were also excluded. The papers published in two journals with the same title, same first author, same study design, sample size, and the same number of in-text citations or references were referred to as duplicate papers. We performed thorough research to identify any such paper and did not include in our review. The principal author’s name was used for reviewing every retrieved research paper for inclusion and exclusion. Moreover, full texts were also retrieved for all such authors until a resolution was reached. The papers included in the research were searched again to find more research articles. The papers that were retrieved again passed through the same screening procedure to ensure and increase the authenticity of the research. 

### 2.3. Step 3: Data Extraction 

A matrix table has been used to display each selected paper. A descriptive analysis was done to review each article. After the descriptive analysis was done, the articles were then compared as Cooper and Koenka [32] suggested. The authors’ names, the study’s objectives, the number of studies with the feedbacks, the identified number of criteria, and the results and feedback were retrieved from the literature reviews. While the analysis was carried out, the data were collected to verify and validate the solutions and results. This analysis was done with the help of identification and description of the issues for every keyword.

### 2.4. Step 4: Evaluate the Quality of Primary Data and Research Synthesis

Detailed information was not available on the antimicrobial properties. The objective of the current study is to collect the data and information regarding the biosynthesis and the antibacterial properties of the Ag-NPs. The content of each article was assessed and evaluated by using a tool, i.e., A Measurement Tool to assess Systematic Reviews (AMSTAR). This tool is particularly used to assess and evaluate the validity and reliability of systematic reviews [33]. In addition, it is also best for the observation [34].

### 2.5. Step 5: Meta-Analyzing and Integrating Their Outcomes of the Synthesis

The selected articles were organized into specific domains of Ag-NPs such as their sizes, shapes, applications, and chemical compositions. The articles regarding the synthesis and functions of silver nanoparticles were grouped together. The articles selected regarding the synthesis of silver nanoparticles were purely based on the silver nanoparticles’ biosynthesis, and they also exhibited effective antibacterial properties. A few steps were additionally taken to analyze the antibacterial properties of the synthesized Ag-NPs. Initially, the author conducted an electronic literature search regarding the silver nanoparticles to find out the original research. The articles selected by the author that documented the biosynthesis of the Ag-NPs were the most relevant, authentic, original, and recently published. Next, the corresponding or first author of the initial article was included, since several articles were present on the antibacterial effects of Ag-NPs. The choice of corresponding or the first author was recommended by Costas and Bordons [35], who are most responsible for preparing the report or developing the idea. After the completion of 5 steps, the review required integration of the results of the reviewed articles, especially to check whether they showed disagreeing findings, and it was required to conduct a second-order meta-analysis before recording the results or findings before the final step. The current study results showed the absence of any discordant results; hence, the meta-analysis was not required. The next section presents the results of the current study. 

## 3. Results

There were about 4327 publications found, the majority of which were from the identified records. A total of 2703 articles were left for testing after the removal of duplicates. After their initial screening, only 83 papers were selected for a full-text examination. Then, their eligibility was checked as per inclusion–exclusion criteria, and 28 articles were selected for the final examination. Figure 1 shows the screening process. During the process, the biosynthesis of silver nanoparticles and antibacterial activities were the main focus. The selected systematic reviews were particularly dealt with the biosynthesis of Ag-NPs through leaves and plants and the antibacterial spectrum they had. Some systematic reviews had also investigated potential antibacterial mechanisms in Ag-NPs- and examined their optimization in orthopedic implants. Some papers were subjected to full-text screening to make decisions quickly; for instance, two studies [36,37] were subjected to screening for their eligibility analysis. The main objective was to find out a methodology that could be used to resolve any disagreement and to achieve consensus. 

Table 1 illustrates the AMSTAR procedure; it shows the lowest score, i.e., 4, and the highest score, i.e., 8. The higher the score is, the better is the quality of the review (e.g., score between 8 and 11 stands for good quality; between 4 and 7 stands for moderate quality; between 0 and 3 read as lower quality) [38]. These scores are based on the 11-item eligibility instrument where the presence of each item in each cited article was given a score of 1 or otherwise 0. Certain citations were categorized as Can’t Answer (CA) or Not Applicable (NA) (See Table 1).

Table 2 shows the 11 items that determined the eligibility criteria of each citation illustrated in Table 1 [38].

### 3.1. Characteristics of Included Studies

The information in Table 3 from the 28 primary studies and reviews seven of which are in vitro studies support that Ag-NPs were biosynthesized successfully. Various biological sources were used for the green synthesis of Ag-NPs, including plants (n = 25), algae (n = 2), and fungi (n = 1). However, the general approach referred to a herbal-mediated fabrication of Ag-NPs (89.28% of studies). These studies utilized different green plants extracts, such as leaf extracts, stem bioresources, and others.

Table 3 comprises 28 primary studies and reviews that support Ag-NPs produced by green synthesis successfully, seven of which are in vitro studies. Various biological sources were used for the green synthesis of AG-NPs, including plants (n = 25, 89%), algae (n = 2, 7.1%), and fungi (n = 1, 3.5%); however, the general approach referred to herbal-mediated fabrication of Ag-NPs (89.28% of studies). 

The majority of these studies contained TEM analysis of nanoparticles that revealed the existence of Ag and confirming the presence of Ag-NPs. These studies have indicated that there are interactions among metallic ions and biomolecules, including proteins, peptides, and amino acids, and some have a major impact on metallic NPs’ therapeutic ability.

### 3.2. Silver Nanoparticles Reduction through Plant Extract as Reducing Agents

The availability of reducing agent in plants, which aid in the synthesis of biocompatible Ag-NPs, was a common aspect of all 28 studies. Secondary metabolites in the extract, such as terpenoids, flavonoids, phenols, alkaloids, and proteins, only operate as reducing agents. These research studies revealed details about such plants, and also the shape and size of plants created through the usage of elemental Ag-NPs This study shows how a flavonoids and phenol compound can be widely utilized for plant extract preparation as a bio-stabilizing agent and bio-reducing agent for the synthesis of zero valent Ag-NPs [50,63]. Proteins may also be used as a bio-reducing agent for silver ions, although they have both benefits and drawbacks [37].

There are also many studies on naturally active ingredients in *Aloe vera* leaves. These include *pectin*, *hemicellulose*, and *lignin*, which can be utilized to lessen Ag ions. These studies show that Ag nanoparticles can be prepared to utilize *Aloe vera* leaves extract as a potential reducing agent. *Aloe vera* leaves reduce Ag salts into Ag nanoparticles, because they contain polyphenols groups [26].

### 3.3. Synthesis

Several factors such as the method used for synthesis, pH, temperature, pressure, time, particle size, pore size, environment, and proximity greatly influence the quality and quantity of the synthesized nanoparticles and their characterization and applications [63]. The reviewed articles also suggest that the bottom–up techniques were predominantly used in the synthesis of Ag-NPs relative to the top–down techniques (<5% of reviewed articles). This is mainly attributed to the surface imperfection of formed particles used in the top–down approach [63]. It should also be noted that while Ag-NPs used a variety of plants and particles of various sizes and shapes, *Azadiractha indica* terrestrial plant leaves and *Sargassum wightii* among them were the most commonly used (29% and 26% respectively). The marine algae (7.1%) were also reported to have been subjected to the synthesis of Ag-NPs antibacterial potential, according to these researchers.

Hence, it is evident that various plants or their extracts are clearly involved throughout the biosynthesis of Ag-NPs of different size and shape and discussed in various publications [30,42]. It has also been established that Ag-NPs can be made in a cubic shape from plant extracts [23,34,35,42] (see Table 3 for details).

Silver nitrate (AgNO_3_) is the most widely used salt precursor accounting for almost 83% of those reported 28 articles using specific synthesis methods. The dominant use of AgNO_3_ is attributed to its low cost and chemical stability when compared to other types of silver salts [64]. The preparation method of Ag nanoparticles utilizing green-reducing agents is also appealing. Extracts of *Azadirachta indica* (Neem), Ocimum *senuiflorum*, *Elephantopus scaber*, and *Carica papaya* (*Papaya*) are also prepared to reduce 1 mM aqueous AgNO_3_ solutions synthesizing Ag nanoparticles.

### 3.4. Size in Relation with Antibacterial Activity

The bactericidal properties of Ag-NPs synthesized from plant extracts have been demonstrated in almost all these 28 studies. This was owing to the ground nature and small scale, which caused the volume ratio to rise upon the surface. The sphere-shaped Ag-NPs are found to be more effective against the *Klebsiella* and *E. coli* bacterial strains than the rod-shaped Ag-NP. The sphere-shaped Ag-NPs exhibit stronger antibacterial activity than rod-shaped and wire-shaped Ag-NPs with similar diameters, suggesting that the shape effect on antibacterial activity is due to the specific and large surface area and facet reactivity, as reported by Raza Maet [65].

Table 4 presents an overview of various elements of different plants and synthesis of their components such as leaf extracts, their shape and size, components used for synthesis, the antibacterial activity and the bacterial impact, and the bio-functionalizing compounds. Other features such as the shape, chemical properties, and size are also mentioned. The particle size of biosynthesized Ag-NPs below 100 nm with various shapes are reported in all these investigations, while the majority of the studies report spherical morphology for biogenic Ag-NPs (60.8%).

According to the data of this work, the average size of Ag-NPs is 30–60 nm, which are spherical in shape along with flavonoid and phenol compounds. All have excellent antibacterial activity against *E. coli* and *S. aureus.*

## 4. Discussion

The findings of this review show that the antimicrobial properties of Ag-NPs and their green synthesis are essential because they are linked to improving human health. Currently, bacterial antibiotic resistance is found to be a significant source of concern in most research studies, the antimicrobial action of NPs being the most effective antibacterial agent. Furthermore, Ag-NPs are used as nanocarriers for antibiotics and drugs, assisting the enhancement of antibiotic activity toward resistant microbes [66]. Several studies have examined plants, having been used as a source of synthesis in many experiments, with benefits such as plant material availability, low price, high availability for mass production, purgative properties, and secondary metabolites. For both the synthesis of Ag-NPs, a correct and controlled need for biological entities would result in well-characterized and highly stable NPs [67]. One of the major benefits of utilizing is identifying the genesis of Ag-NPs existing in plants and discovering the non-toxicity and metabolites causing silver reduction.

### 4.1. Ag-NPs Application and Mechanism of Action

Previous studies have attributed Ag-NPs antibacterial activity to the release of silver ions, which can be produced and introduced by AG-NP oxidative dissolution in the presence of oxygen [68,69]. Similarly, silver ions strongly attract electron-donating groups such as sulfhydryl, amino, imidazole, phosphate, and carbonyl groups found on membranes and proteins [70,71]. Silver ions can form persistent AS-Ag bonds with protein thiol groups (ASH), altering the 3D structure of proteins and blocking active binding sites [72]. As a result, silver ions can prevent the movement and release of potassium (K+) ions from microbial cells, as well as the creation of adenosine triphosphate (ATP) [73]. Furthermore, silver ions may easily combine with various biomolecules, including DNA, RNA, and peptides, generating insoluble complexes that prevent cell division and reproduction [74,75]. 

Possible mechanisms of action of AG-NP are shown in Figure 2 and Figure 3 [76,77]. Figure 2 illustrates the action mechanism of Ag-NP against bacterial cells, activities of which include membrane damage in the form of inhibition of cell multiplication or Reactive Oxidative Species (ROS) formation seen as cell damage or destruction. The former action of inhibiting the cell multiplication is due to the association of Ag-NP with DNA or other biomolecules, while the latter action of cell destruction is a result of the interaction of enzymes and molecules.

Figure 3 summarizes how AG-NP is attached to cell membranes, membrane proteins, and DNA bases, disrupting normal function (blue arrows. This generates silver ions and affects membranes, DNA, and proteins (red arrows. This accumulates reactive oxidative species (ROS), which may also affect DNA, cell membranes, and membrane proteins (black dotted line) [77].

### 4.2. ROS-Based Antibacterial Effects of Silver Ions

The specific mechanism of silver nanoparticles’ antibacterial properties can be further studied through various antibacterial actions It is well-known that Ag-NPs continuously discharge silver ions, which are a potential microbe-killing mechanism [78]. These silver ions can adhere to the cell wall and cytoplasmic membrane due to electrostatic attraction and affinity to sulfur proteins. The adhered ions can improve the cytoplasmic membrane permeability and result in disruption of the bacterial envelope [79]. The respiratory enzymes can also be disabled once free silver ions enter into cells, resulting in reactive oxygen species (ROS) but no ATP synthesis results [80]. ROS however can trigger cell membrane rupture and DNA alteration. Since sulfur and phosphorus are key components of DNA, the interaction of silver ions with these elements can create issues with DNA replication, cell reproduction, and even microorganism death. Furthermore, silver ions can prevent protein production by denaturing ribosomes in the cytoplasm [81]. 

Figure 4 exhibits a few antibacterial actions of silver nanoparticles (Ag-NPs) such as Cell wall and cytoplasmic membrane are disrupted and silver nanoparticles release silver ions (Ag+), which attach to or pass through the cell wall and cytoplasmic membrane. Ribosome denaturation such as silver ions also denature ribosomes, inhibiting protein synthesis. Inhibition of adenosine triphosphate (ATP) production too results in silver ions deactivating respiratory enzymes in the cytoplasmic membrane, preventing ATP production. Membrane disruption is also seen to be caused by reactive oxygen species (ROS) produced by a disrupted electron transport chain: that is, ROS produced by a broken electron transport chain can induce membrane disruption. It is also seen that silver and reactive oxygen species bind to deoxyribonucleic acid (DNA) and hinder replication and cell proliferation while silver nanoparticles can be seen accumulating in cell wall pits, causing membrane denaturation. As a result, membrane perforation is caused and silver nanoparticles penetrate the cytoplasmic membrane directly and release organelles from the cell. These are evidence of the antibacterial actions of silver nanoparticles (Ag-NPs), causing disruption and its variants within a cell.

Plant extracts are favored due to having a slow rate of nanoparticle synthesis using microbes as well as their simplicity, performance, and feasibility. Plants’ capacity to absorb and detoxify heavy metals has been well established [82,83]. Furthermore, Ag-NPs are made from plants and could be easily prepared from plant extracts or perhaps even from the entire plant [84]. Plants containing reducing and stabilizing agents are of a great help in the development of biocompatible Ag-NPs as well as secondary metabolites inside the extracts such as terpenoids, flavonoids, phenols, alkaloids, proteins, and carbohydrates, which often function as reducing agents. [36,48,58,85] discover that for the production of silver nanoparticles using plant extracts, *Thymus kotschyanus* leaf extract was specifically used to synthesize Ag-NPs. The spherical Ag-NP particles were obtained with the size varied in the range of 50–60 nm. Later, a group of researchers [63] made use of spherical silver nanoparticles using different extracts viz., Juglans *regia* (bark), seaweed *Ulva flexuosa*, and *Phoenix dactylifera* with sizes varying from 15 to 60 nm.

Similar results were achieved by [44,45]. in which *Acacia rigidula* and *Azadirachta indica* leaves extract were used to synthesize Ag-NPs, with shape being spherical and the size varying in the range 8–60 nm and 48 nm. The spherical silver nanoparticles were also utilized in other studies [47] using *Ocimum senuiflorum* extract with sizes from 25 to 40 nm and Asmaa [48] who utilized *Pelargonium graveolens* of size ranging from 16 to 40 nm. Das [63] used Ag-NP in cubic form from plant extracts to prepare elemental silver nanoparticles using the leaf extract of *Banana peel* and *Aloe vera*, with the synthesized Ag-NPs varying in size of about 23 nm and 15 nm, respectively.

Ag-NPs made from natural extracts with a shape other than spherical have been reported in a very limited number of articles [57]. Seven studies found a cubic shape, and two others examined the synthesis of trigonal Ag-NPs with a cubical form [86,87]. Only one study [88] used *Acorus calamus* extracts to classify the spherical shapes of Ag-NPs with a size of 83 nm. Most of these studies utilized synthesized nanoparticles spherical with a shape of less than 100 nm. Each of these studies has also reported strong antibacterial activity toward pathogenic microorganisms including *Staphylococcus* and *Pseudomonas*. The antimicrobial activity of the synthesized Ag-NPs has also been significantly reported against the growth of *Proteus vulgaris*, *Escherichia coli*, and *Klebsiella pneumoniae* strains with different concentrations while treating *Fagonia cretica* that were spherical in shape [51].

### 4.3. In Vitro Cytotoxicity

A phenomenon noticed in a few studies was the discussion of toxic effects of Ag-NPs on different cell lines, including macrophages (RAW 264.7), bronchial epithelial cells (BEAS-2B), alveolar epithelial cells (A549), hepatocytes (C3A, HepG2), colon cells (Caco2), skin keratinocytes (HaCaT), human epidermal keratinocytes (HEKs), erythrocytes, neuroblastoma cells, embryonic kidney cells (HEK293T), porcine kidney cells (Pk 15), monocytic cells (THP-1), and stem cells [22,89]. The exposure of A549 cells was also found in increasing concentrations of Ag-NPs for 24 h and causes morphological changes such as cell shrinkage, cellular extensions, a specific spreading pattern, and cell death in a dose-dependent manner [58,63]. The results of in vitro studies further indicate that Ag-NPs are toxic to the mammalian cells that are derived from the skin, the liver, the lung, the brain, the vascular system, and reproductive organs [36]. The cytotoxicity of Ag-NPs depends on their size, shape, surface charge, coating/capping agent, dosage, oxidation state, agglomeration, and type of pathogens against which their toxicity is investigated [52,63]. Despite these studies, the toxicological nature of Ag-NPs mechanism is still unclear.

### 4.4. Ag-NPs Application

One of the most significant advantages of nano silver-based biomaterials is their inherent anti-pathogenic properties, which can be seen in both planktonic and biofilm-organized microorganisms. Antimicrobial peptides such as polymyxin B are used to functionalize nanoparticles and functionalizing molecules such as sodium borohydride, which are used to create effective biocompatible Ag-NPs with high penetration ability [63]. There is also a lot of factual information about Ag-bio-NP’s applications [90,91,92], such as that there is the chance of Ag-NPs having antimicrobial activity, as evidenced by the information reported [30,35,42].

Due to the large surface area and small nano-size, the processed zirconia nanoparticles show strong antibacterial activity against *Salmpnella typhi*, *Bacillus subtilis*, and *Escherichia coli*. Researchers also reveal the antibacterial activity of biogenic silver nanoparticles where biosynthesized Ag-NPs and use of the leaf extract of a plant *Protium serratum* are reported in detail [93]. These reports show that the formulated Ag-NPs displayed strong antibacterial activity toward food-borne pathogens such as *Pseudomonas aeruginosa*, *Escherichia coli*, and *Bacillus subtilis* [94,95]. One of the first papers in this list examined a leaf extract of *Cucumis sativus* to make elemental Ag-NPs. Using extracts of the leaf of *Ocimum tenuiflorum* for “green synthesis,” the researchers [96] observed the presence of prismatic silver nano peptides whose size range from 25 to 40 mm. Another research [97] used *Capsicum annuum* from 50 to 70 nm to make extracts with spherical-sized silver nanoparticles. Singh [66] suggested that *Cannabis sativa* can also be used to make Ag-NPs, with extract of leaf of *Argemone mexicana* acting as a capping and reducing agent when added to an aqueous solution of silver nitrate. These findings confirm that silver nanoparticles may have antibacterial activity against E. coli [46].

Ag-NPs made from natural extracts having other than spheroid shapes have been reported in a very limited number of articles. The synthesis of trigonal Ag-NPs with a cubical form has been examined in two articles [91,92]. One article used Acorus calamus extracts to classify the Ag-NPs having spherical shapes and sizes of 83 nm [93]. The synthesized nanoparticles have demonstrated powerful antimicrobial activity toward pathogenic microorganisms such as *Staphylococcus* and *Pseudomonas* in various other research papers. UV-Vis spectrometer, X-ray diffractometer (XRD), scanning electron microscopy (SEM), and Fourier Transmission Infrared (FTIR) spectrophotometer are also used to investigate the properties of NPs. It is emphasized that geometric forms such as oval or spherical structures are conglomerates of the smaller embryonic elemental silver particles [98]. Higher-resolution images of spherical silver nanoparticles obtained from SEM would further authenticate it [99]

Hence, in this review, we intended to provide a brief overview of Ag-NP biosynthesis from plant extracts in order to investigate their antibacterial capacity. It was premised that Ag-NPs were commonly used during electronics and photonics as biosensors in biocatalysis, protein coagulation, and drug delivery due to their oxidation stability. Silver nanoparticles have been used to coat cutlery, door handles, and a computer keyboard and mouse; these are also used to make new coatings and cosmetics and air-conditioning filters, lakes, toilets, and other areas [57].

The environmental and economic problems associated with most Ag-NPs synthesis methods have led to a search for many other environmentally and economically beneficial options. Usually, due to its multiple health, economic, financial, and medicinal advantages, the biological process of synthesis utilizing plant resources has been considered suitable for the production of Ag-NPs. For example, Ag-NPs synthesized with plant extracts have such unique properties that have also boosted their use in agriculture for fertilizers, pesticides, and fumigants. Phyto synthesized Ag-NPs have found widespread use in the manufacture of disinfectant, antifungal, anticancer, antioxidant, anti-inflammatory, and antidiabetic agents in medicine and pharmacy [100]. Other plant extracts (as raw material for Ag-NPs synthesis) of *Brassica oleracea*, *Brassica oleracea*, *Ocimum tenuiflorum*, and *Cola nitida* pod plants are also suggested based on findings. These extracts have a substantial impact on *E. coli*, *Pseudomonas*, and other Gram-negative rods. Leaf extracts are also given preference during the synthesis of Ag-NPs, which can be seen in Table 2, although some sections of related plants—flowers, seeds, fruits, and so on—could also be used in some instances.

## 5. Limitations of the Study

There are a few limitations to conduct a literature review of this nature. For instance, it was difficult to extrapolate the results directly of such experimental studies to a clinical setting due to skills and instruments deficiency. The testing of both clinical and standard reference strains leads to achieving more realistic results. Although 28 (n) research papers were reviewed systematically in view of study objectives, simultaneous characteristics (e.g., application of silver nanoparticles, anti-biofilm, antibacterial activity) in each one of them were not present. Therefore, the variation of the number of included studies for all of these characteristics was different. However, despite these limitations, this study will serve as a good source of reference for finding relevant studies and building a point of view on silver nanoparticles and their applications in preventing biofilm formations.

## 6. Conclusions

The analysis focuses on research that has been done mostly on the biosynthesis of Ag-NPs utilizing different plant extracts, as well as the elimination utilizing biological components. Biogenic NPs have been shown to have a significant antibacterial effect for all or the majority of the bacteria, such as *Proteus vulgaris*, *Pseudomonas aeruginosa*, *Staphylococcus aureus*, *Vibrio cholera*, and *Klebsiella* spp. While biosynthesis becomes less toxic or environmentally sustainable, reducing compounds necessitates more research to comprehend surface chemistry fully. A review of many studies confirmed this trend. Furthermore, Ag-NPs, made from a variety of plants and algae, has a powerful antibacterial ability. This area has yielded several studies. We hope that this approach of green synthesis will aid researchers in determining the long-term advantages of Ag-NPs generated by biosynthesis in their various potential applications. Ag-NPs are appealing because they are harmless to humans at low concentrations and have antibacterial activity throughout a broad spectrum. Nevertheless, before applying the applications of Ag-NPs, the potential toxicological effects must be investigated.

## Figures and Tables

**Figure 1 molecules-26-05057-f001:**
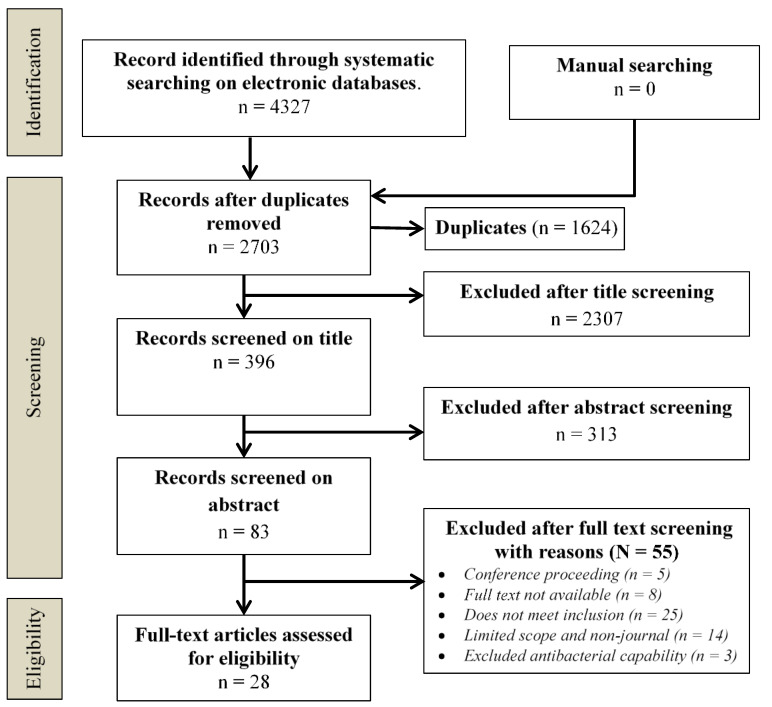
PRISMA (Preferred Reporting Items for Systematic Reviews and Meta-Analysis).

**Figure 2 molecules-26-05057-f002:**
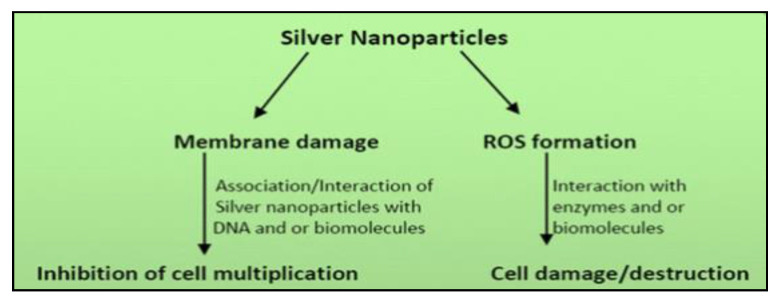
Mechanism of action of silver nanoparticles against bacterial cells [76].

**Figure 3 molecules-26-05057-f003:**
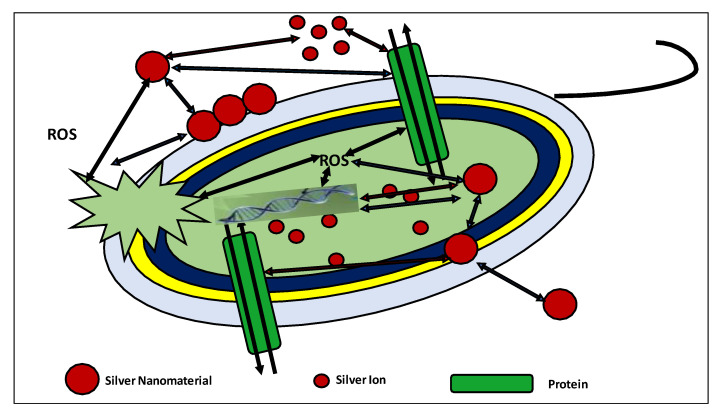
Segregation of AG-NPs silver ions and proteins disrupting the general funciton of the cell [77].

**Figure 4 molecules-26-05057-f004:**
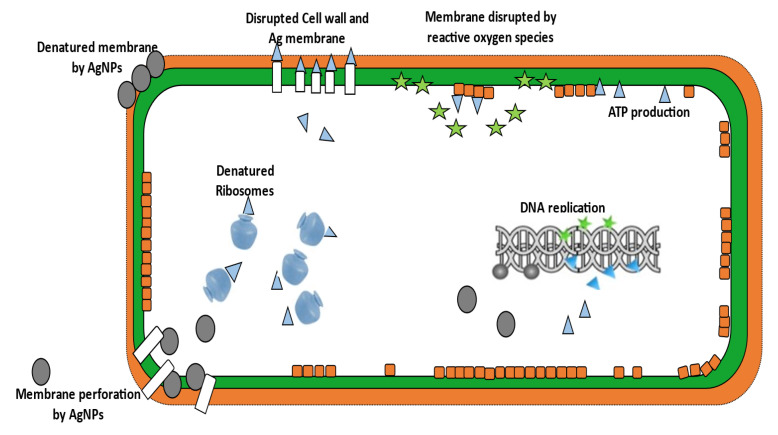
Antibacterial actions of silver nanoparticles (Ag-NPs).

**Table 1 molecules-26-05057-t001:** AMSTAR table.

		Quality Assessment Tool AMSTAR											
	Citation	1	2	3	4	5	6	7	8	9	10	11	Score
1	Khan et al. 2018 [39]	1	CA	1	1	0	1	NA	1	1	1	1	8
2	Ahmed et al.2019 [40]	1	0	1	1	0	1	NA	1	1	1	1	8
3	Mishra et al. 2019 [41]	1	0	0	1	0	1	NA	1	1	1	1	8
4	Roy et al. 2019 [42]	1	0	1	1	0	1	1	1	NA	1	1	8
5	Nasrollahzadeh et al. 2019 [43]	1	0	1	1	0	1	1	1	NA	1	1	8
6	Zafar et al. 2019 [44]	1	0	1	1	0	1	NA	1	NA	1	1	7
7	Nisar et al. 2019 [45]	1	0	1	1	0	1	NA	1	NA	1	1	7
8	Some et al. 2018 [46]	1	0	1	CA	0	1	NA	1	NA	1	1	6
9	Fahimirad et al. 2019 [47]	1	0	1	CA	0	1	NA	1	NA	1	1	7
10	ElShafey 2020 [48]	1	0	1	CA	0	1	NA	1	1	0	1	6
11	Kumar et al. 2018 [49]	1	0	1	CA	0	1	NA	0	NA	1	1	5
12	Singh et al. 2020 [37]	1	0	1	1	0	1	NA	1	NA	1	1	7
13	Salleh et al. 2020 [50]	1	0	1	1	0	1	NA	1	NA	0	1	6
14	Yun’an Qing et al. 2018 [36]	1	CA	1	1	0	1	NA	1	NA	CA	1	6
15	Ferdous 2020 [51]	1	CA	1	1	0	1	NA	1	NA	0	1	6
16	Yin et al. 2020 [52]	1	CA	1	1	0	1	NA	1	1	1	1	6
17	Ahmad et al. 2019 [53]	1	0	1	1	0	1	NA	1	NA	1	1	7
18	Gumel et al. 2019 [25]	1	0	1	1	CA	1	NA	1	NA	1	1	7
19	Escárcega-González et al. 2018 [54]	1	0	1	0	0	1	NA	1	NA	1	1	6
20	Nagar et al. 2018 [55]	1	CA	1	0	0	1	NA	1	NA	0	1	5
21	Mikhailov et al. 2018 [56]	1	0	1	1	0	1	NA	0	NA	1	1	6
22	Hamelian et al 2018 [57]	1	0	0	0	0	1	NA	1	NA	0	1	4
23	Zulfiqar et al. 2018 [58]	1	0	0	1	0	0	NA	0	NA	1	1	4
24	Haqq 2018 [59]	1	0	0	0	0	1	NA	0	NA	1	1	4
25	Ishak et al. 2019 [60]	1	0	CA	0	0	1	NA	1	NA	1	1	5
26	de Aragao et al. 2019 [61]	1	0	CA	0	0	1	NA	1	NA	1	1	5
27	Hasnain Met al. 2019 [62]	1	0	CA	1	0	1	NA	1	NA	1	1	6
28	Das et al. 2020 [63]	1	0	CA	1	0	1	NA	1	NA	1	1	6

1 = Yes; 0 = No; CA = Can’t Answer; NA = Not Applicable (Source: Sharifetal. 2013) [38] Characteristics of Included Studies.

**Table 2 molecules-26-05057-t002:** Eligibility criteria of citations.

S. No	Eligibility Item
1.	Provided with the prior design.
2.	Extraction of data and selection of duplicate study is made.
3.	Literature has been searched comprehensively.
4.	The publication has passed the inclusion criteria.
5.	An index of articles (with inclusion and/or exclusion) is given.
6.	Features of included articles are provided.
7.	The scientific quality of the included studies was evaluated before documenting.
8.	The scientific quality of the included studies was used to formulate conclusions.
9.	Appropriate methods were used to combine the studies’ findings.
10.	The likelihood of publication bias was assessed.
11.	The conflict of interest was stated.

**Table 3 molecules-26-05057-t003:** Characteristics and observations from the systematic reviews.

Author	Year	Objective of Study	Summary Finding
Yun’an Qing et al. [36]	2018	To determine the potential antibacterial mechanisms of Ag-NPs. To elaborate methods to enhance the biocompatibility of Ag-NPs.	To avoid implant-related infection and show how Ag-NPs with high antibacterial efficacy are commonly used in implant surface modification.
Ferdous [51]	2020	To elucidate the factors such as the size, shape scale, surface chemistry, and stability. To examine how Ag-NPs’ antibacterial activities are influenced by structural factors, which could aid in the development of more effective Ag-NPs.	How the defined structural factors such as size, shape scale, surface chemistry, and stability affect the antibacterial mechanism of Ag-NPs.
Yin et al. [52]	2020	To gather the most up-to-date information on the biomedical applications of Ag-NP-based nanostructures.	It is centered on the recent data on Ag-NP-based nanostructures’ biomedical applications, and parameters such as toxicity, physiochemical, and bio-functional properties,
Ahmad et al. [53]	2019	To assess the green synthesis, characterization, and biological activities of Ag-NPs using a variety of biological sources.	In the field of nanotechnology, green synthesized Ag-NPs have unrivaled significance. Ag-NPs have a broad range of pharmacological operations, and their cost-effectiveness makes them a viable alternative to local medicines.
Hamelian et al. [57]	2018	To focus on Thymus-based green silver nanoparticle synthesis. To investigate an antibacterial, antioxidant, and cytotoxic effects of synthesized nanoparticles.	Thymus Kotschyanus extract was used in this study to synthesize Ag-NPs in an environmentally friendly, healthy, and practical way. There were no chemical substances involved. Silver nanoparticles with a diameter of 50 nm in this herb have a strong antibacterial and antioxidant impact.
Gumel et al. [26]	2019	To learn more about silver nanoparticle biogenesis and the mechanisms that underpin their antimicrobial efficacy.	The antimicrobial properties of silver nanoparticles and plant extracts, such as antibacterial and antifungal properties, are demonstrated in this report.
Escárcega-González et al. [54]	2018	To develop a green one-pot synthesis process for Ag-NP production that incorporates the *Acacia rigidula* extract as a therapeutic agent to treat pathogens.	The results show that the Ag-NPs used in this study can destroy pathogenic bacteria.
Nagar et al. [55]	2018	To investigate whether a leaf broth of *A. indica* can be used as a reducing and capping agent to synthesize Ag-NPs.	The biosynthesized Ag-NPs is classified using a variety of instrumental techniques. The particles were described as crystalline average size cubical particles with a high level of stability.
Ahmad et al. [40]	2019	This analysis focuses on the synthesis of biological MNPs by plants and microbes, as well as their cellular uptake, biocompatibility, cytotoxicity, and biomedical applications.	The synthesis of MNPs is influenced by temperature, incubation time, and pH. This study found that biologically synthesized MNPs had higher biocompatibility than MNPs synthesized using different physicochemical methods.
Mishra et al. [41]	2019	The aim of this research was to look into current Ag-NP biosynthesis trends; and To find out if they have antimicrobial activity and any biotechnological potential as well.	Ag-NPs are regarded as a crucial expansion in the continuum of nanomaterials due to the versatile qualities it offers in terms of application in various fields of study. It is likely that NP synthesis will be used in the future to make antimicrobial compounds in biomedical nanotechnology.
Mikhailov et al. [56]	2018	To synthesize Ag-NP using different physicochemical methods. To look at a certain synthetic method that use biological objects to make elemental silver nanoparticles.	At this time, experimentally determining the scale, form, etc. and feasibility of biosynthesized Ag-NP dispersion. Implementing silver nanoparticles NP biosynthesis and predefined parameters will eventually necessitate the development of new concepts and methods.
Roy et al. [42]	2019	To address recent developments in green synthesis of silver nanoparticles, while the mechanism of antimicrobial action underpins their use as antimicrobial agents.	Nanoparticles appear to be able to cross the membrane and cause damage. Loading drugs on the nanoparticle surface can increase the efficiency of biocidal motion in addition to disrupting the membrane.
Zulfiqar et al. [58]	2019	To determine if plant extract *Fagonia cretica* could be used as a reducing and stabilizing agent in the synthesis of Ag-NPs and to see how effective the extract is against bacteria.	According to morphological and structural characteristics, the study found Ag-NPs as hugely crystalline, averaging 16 nm size, and the presence of active bio-reducing and stabilizing agents in the *Fagonia cretica* extract. Ag-NPs revealed antibacterial activity against a few other plant extracts
Nasrollahzadeh et al. [43]	2019	To examine whether by reducing Ag+ ions (plant extracts) and controlling the size of the NPs, the Ag-based nanoparticles can be produced.	This research looked into the green synthesis of Ag-based nano catalysts such as Ag-NPs, AgPD NPs, and AU Ag-NPs.
Zafar et al. [44]	2019	To emphasize the importance of plant extracts in the bio-fabrication of nanoparticles as a renewable, non-toxic, and environmentally friendly process.	Ag-NPs have been shown to be effective in treating *M. Incognita*. Silver nanoparticles are used in food packaging to increase the shelf life of the product.
Nisar et al. [45]	2019	To assess the antimicrobial properties of various biosynthesized metal nanoparticles, as well as the mechanisms by which they work.	Several antimicrobial green-base nanoparticles have been successfully developed from a variety of biological sources, the most prominent of which are plants. These bio-nanomaterials have proved to be effective against bacterial and fungi that cause disease (both plant and human).
Some et al. [46]	2019	The goal of this work is to look into the synthesis and characterization of biomolecule-capped Ag-NPs; and To evaluate antimicrobial properties in the presence of human and plant pathogens.	Biomolecules act as both reducing and stabilizing agents in the green pathway, resulting in biocompatible NPs. In the literature, promising findings on Ag-NPs’ antimicrobial activity against a variety of pathogenic microorganisms have been recorded.
Haqq et al. [59]	2018	To examine the biological activities of Ag-NP and plant-mediated green synthesis.	The potential of Ag-NPs to perform in a number of bioassays has also been lauded. This study would help researchers create new Ag-NP-based drugs using green technology.
Ishak et al. 2019 [60]	2019	To share the latest research on metal and metal oxide nanoparticles, including silver and gold nanoparticles; and to issue directions and implementations for green synthesis methods based on plant extracts.	Plant extracts have attracted a lot of interest because of their ability to minimize and stabilize metal nanoparticles in a single phase using their unique natural properties.
El Shafey [48]	2020	To focus on MNP and Monps biosynthesis procedures, including a comparison of green synthesis and conventional chemistry methods, as well as several new directions for green synthesis of nanoparticles from various plant parts, particularly plant leaf extract.	The environmentally sustainable and general approach can be extended to a number of therapeutic and scientific uses, as well as other noble metals such as Ag and Pd. The low cost and ease of synthesis of antimicrobial nanoparticles using local plant extracts without the use of a toxic chemical reducer are the main advantages of the greener preparation methods.
de Aragao et al. [61]	2019	To make silver nanoparticles, researchers use a natural polysaccharide derived from red marine algae (*Gracilaria birdiae*). To monitor the antimicrobial activity of the synthesized NPs against representative strains of *Staphylococcus aureus* and *Escherichia coli*.	Ag-NPs were tested for antimicrobial activity against Gram-negative and Gram-positive *Escherichia coli* and *Staphylococcus aureus*, and both samples showed antimicrobial activity against *E. coli*. The Ag-NPs were made using natural sources such as red algae, which have favorable properties, in a simple, fast, and one-step process.
Hasnain et al. 2019 [62]	2019	Synthesis of stability in silver nanoparticles from extract of purple heart plant leaves using a biological reduction technique. To see if it is successful against bacteria.	According to antibacterial activity testing, purple heart plant extracts primarily resulted in the removal of silver ions and the stabilization of silver nanoparticles. These purple heart plant leaves extract-mediated synthesized silver nanoparticles have antibacterial activity against *E. coli* and *S. aureus* at a concentration of 100 µg/mL, which is much better than the extract concentration.
Khan et al. [39]	2018	To provide basic information about medicinal plants and silver nanoparticles and show whether they have antiviral, bactericidal, and fungicidal properties. To demonstrate how medicinal plants can be used in a wide range of applications.	Promising non-chemicals have no effect on adult bees (plant extracts). This study attempted to determine the current state of medicinal plant science worldwide.
Kumar et al. [49]	2019	The aim of this study was to plan, classify, and evaluate the potential of G-Ag-NPs as a wound treatment against human pathogenic bacteria.	According to the current study, the novel G-Ag-NPs demonstrated strong antibacterial properties against both Gram-negative and Gram-positive bacterial strains, suggesting that they have a lot of potential for treating pathogen-infected wounds.
Fahimirad et al. [47]		To gain a thorough understanding of the Ag-NPs synthesis process as it is mediated by plants. To evaluate antimicrobial and cytotoxic properties, as well as their implementations.	These nanoparticles were found to be non-toxic to normal human cells at therapeutic concentrations.
Singh et al. [36]	2020	To research into the green synthesis of different metal NPs that have been verified. To assess the antibacterial properties’ different modes and mechanisms.	Nanoparticles interact with DNA, enzymes, ribosomes, and lysosomes, influencing cell membrane permeability, oxidative stress, gene expression, protein activation, and enzyme activation.
Das et al. [63]	2020	To discuss the synthesis of Ag-NPs by plants and algae, as well as their use as an antimicrobial agent.	Ag-NP biosynthesis has been investigated using a number of plants and algae, as well as Ag-NP reduction using biological components.
Salleh et al. [50]	2020	To learn more about the mechanisms that cause Ag-NPs to have antiviral and antibacterial effects on microorganisms.	Ag-NPs’ specific physicochemical properties are influenced by a variety of factors, including scale, surfactant, and structure morphology.

**Table 4 molecules-26-05057-t004:** Plant extracts that were used for biosynthesis of silver nanoparticles.

Plant	Used Component for the Synthesis	Size (nm)	Shape	Bacterial Impact	Bio-Functionalizing Compounds	Reference
*Thymus kotschyanus*	Extract	50–60	Spherical	*Escherichia coli* and *Staphylococcus aureus*	Protein	[58]
*Juglans regia (Bark)*	Leaf	15–30	Cubic and smooth	*Streptococcus mutans*	Flavonoids	[63]
*Seaweed ulva flexuosa*	Extract	42–83	Spherical	*Antibacterial action against Escherichia coli*, *Staphylococcus aureus*	Protein	[44]
*Acacia rigidula*	Extract	8–60	Spherical	*Escherichia coli*, *Bacillus subtilis*, *P. aeruginosa*	Phenol compound	[45]
*Azadirachta indica*	Leaf	48	Cubic	*E. coli*	Protein	[46]
*Ocimum tenuiflorum*	Extract	25–40	Linear	*E. coli* and *B. subtilis*	Flavonoids	[47]
*Elephantopus scaber*	Extract	37	Spherical	*B. subtilis*, *L. lactis*, *P. aeruginosa*, *A. penicillioides*	Proteins	[50]
*Fagonia cretica*	Extract	16	Spherical	*Proteus vulgaris*, *Escherichia coli* and *Klebsiella Pneumoniae*	Hydroxyl and secondary amines	[51]
*Carica papaya (Papaya)*	Leaf	60–80	Spherical	*Escherichia coli*, *Staphylococcus aureus*	Proteins	[52]
*Argemone mexicana*	Leaf	30	Cubic	*E. coli* and *B. subtilis*	Leaf proteins and metabolites	[53]
*Datura stramonium*	Leaf	15–20	Spherical	*Streptococcus mutans*	Flavonoids, terpenoids	[54]
*Cola nitida pod*	Extract	12–80	Cubic	*E. coli*, *P. aeruginosa* and *Klebsiella*	Protein	[52]
*Taraxacum officinale*	Leaf	15	Cubic and hexagonal	*Xanthomonas axonopodis* and *P. syringae*	Flavonoids, terpenoids, and triterpenes	[26]
*Rosa indica*	Leaf	1–100	spherical	*P. aeruginosa* and *Bacillus subtilis*	Polyphenol	[55]
*Phoenix dactylifera*		15–40	Cubic	*E. coli*	Polyphenols, lipids, and fatty acids	[53]
*Mangosteen*	Extract	30	Spherical	*E. coli* and *S. aureus*	Flavonoids	[58]
*Rheum palmatum root*	Extract	121	Cubic	*S. aureus* and *P. aeruginosa*	Flavonoids, terpenoids	[64]
*Prunus japonica*	Extract	26	Spherical	*Proteus vulgaris*	Protein	[59]
*Boerhaaviadiffusa*		25		*F. ranchiophilum*	Phenol	[60]
*Banana peel*	Extract	23.7	Cubic	*E. coli*, *P. aeruginosa*	Lips and fatty acids	[61]
*Aloe vera*	leaf	15.2	Cubic	*S. aureus*	Protein	[62]
*Pelargonium graveolens* *(Geranium)*	Leaves	16–40	Spherical	*E. coli*, *P. aeruginosa*	Flavonoids	[63]
*Sargassum wightii*	Extract	68.04	Cubic	*S. aureus* and *P. aeruginosa*	Protein	[26]

## Data Availability

No new data were created or analyzed in this study. Data sharing is not applicable.

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
