# Peer review of "Systematic Review on Biosynthesis of Silver Nanoparticles and Antibacterial Activities: Application and Theoretical Perspectives"

_molecules, 2021, doi:10.3390/molecules26165057_

Round 1

Reviewer 1 Report

This manuscript is the second version I have seen. I think the author has made careful revisions in accordance with the review comments. I recommend accepting this manuscript.

Author Response

Thank you for your kind review, appreciation and motivation.

Regards

Reviewer 2 Report

The authors wrote a Systematic Review on Biosynthesis of Silver Nanoparticles and Antibacterial Activities. I congratulate them for a very good job done. The article is well written and organized, comprehensive, critical having a structure that covers a lot of aspects in the field. I like especially the way how they explained the literature searching process, inclusion / exclusion criteria, and data extraction. As a result of my evaluation, I have some minor comments only.

Line 50-51 is it necessary to mention (Lee & Jun, 2019), line 378 (Ramkumar et al., 2017), line 3838 (Durán et al., 2016), once you have the reference?

Please point out the originality of the article.

AgNO3 should be AgNO3 –please check in the whole text

The caption of Figure 4 is quite long. The authors could consider explaining the mechanism action in the text and keeping the caption shorter.

Line 416 – I would not start a sentence with a number (7). In my opinion is better to simply write “Seven”

The next two references (    DOI: 10.1016/j.ab.2019.05.007 and     DOI: 10.1088/1752-7163/aa820f) can be included in the subheading “Ag-NPs Application”

Author Response

Thanks, sir .I uploaded file.

This manuscript is a resubmission of an earlier submission. The following is a list of the peer review reports and author responses from that submission.

Round 1

Reviewer 1 Report

This review paper analyzes and sorts out the research status of silver nanoparticles in the field of biosynthesis and antibacterial properties through the method of big data analysis. The method of this paper is innovative and the study is meaningful. It is suggested it can be accepted after the following issues are resolved.

  1. Some important relevant papers were not included in the introduction section, when the authors mentioned the state of art on antibacterial activities of Ag nanoparticles.
  • Controlled synthesis of Ag nanoparticles with different morphologies and their antibacterial properties. Sci. Eng. C, 2013, 33(1): 397-404.
  • Synthesis of poly acrylic acid modified silver nanoparticles and their antimicrobial activities. Materials Science and Engineering: C, 2014, 41: 249-254.
  • A Facile Method to Prepare Size-tunable Silver Nanoparticles and Its Antibacterial Mechanism. Advanced Powder Technology, 2018, 29(2): 407-415.
  • Synthesis of silver nanoparticle-decorated hydroxyapatite (HA@Ag) poriferous nanocomposites and the study of their antibacterial activities. RSC Advances, 2018, 8(73): 41722-41730.

  1. The green method to synthesize Ag nanoparticles has advantages, but there are also some problems. For example, the purity of the sample is low, the by-products are rich, and the preparation method is difficult to scale. These should be explained in the manuscript.

  1. Some text needs to be revised, e.g. P3 Line106, AG-NP.

  1. P4, Line 178: How to remove the duplicated papers (1624)? The authors should explain it in detail.

  1. According to the author's literature research, does the preparation method of Ag nanoparticles prepared by green synthesis have any effect on their antibacterial properties? It has been reported that the smaller the nanoparticles, the better the antibacterial properties. Can the green synthesis method obtain monodisperse small-size silver nanoparticles?

  1. The author needs to clarify the future development trend of the field.

  1. There are too few illustrations in the text and the tables are too long, which affects readers’ reading experience. It is recommended to add 1-2 general and informative illustrations.

Reviewer 2 Report

Discussion part:

  1. Description of mechanism for antibacterial effects of the silver ions should be included with additional Figures or Schemes.
  2. Description of detail mechanisms for ROS-based antibacterial effects of silver ions to disrupt synthesis of ATP, cell death and apoptosis is strongly recommended to be added with additional Figures and recent References.

Reviewer 3 Report

Extensive editing of English language, style, and sometimes grammar is required as throughout the manuscript there are some incomprehensible phrases and sentences. I recommend the Authors reorganizing and correcting the manuscript. Then, an English native speaker with scientific background, to remove spelling inaccuracies and to make the manuscript formal, should revise the manuscript.

The review should be totally reorganized giving consistency and being careful to avoid useless phrases and to emphasize the key sentences.

Please consider some specific comments and suggestions. I would like to point out that this list is not exhaustive and represents only few examples.

  • There are many phrases that should be reorganized because the form or/and the style are not correct or because are not written in a clear and consistent way. For example:

Line 40-43 abstract

Line 62-64 and 66-68 introduction

Line 39-42 discussion

Line 48-50 discussion

  • There are many words to correct, for example:

“In-vitro students” (in-vitro test?)

“Sifferent” (different?)

“Living things” (living being?)

  • There are words that should be changed because are not formal or explained, for example:

“An emphasis is laid”

“Wong suggest biological methods…” Which biological methods?

“other biological methods” Which other biological methods?

“purgative proprierties”

“Silver ions consumed”

“Made in this manner”

“Ag-NPs can also collect on”

Please, write the microorganisms’ name with cursive style.

  • Some grammar mistakes.

“it lead”

“Ag-MPs gaining” (are gaining)

“Wong et al (2010) suggest” (suggested is better)